# Damage of Hygrothermally Conditioned Carbon Epoxy Composites under High-Velocity Impact

**DOI:** 10.3390/ma11122525

**Published:** 2018-12-12

**Authors:** Xiang Liu, Weimin Gu, Qiwen Liu, Xin Lai, Lisheng Liu

**Affiliations:** 1Department of engineering structure and mechanics, Wuhan University of Technology, Wuhan 430070, China; liuxiang1129@whut.edu.cn (X.L.); gwmwhut@sina.com (W.G.); laixin@whut.edu.cn (X.L.); 2Hubei Key Laboratory of Theory and Application of Advanced Materials Mechanics, Wuhan University of Technology, Wuhan 430070, China; 3Faw jiefang automobile Co. Ltd, Changchun 130011, China; 4State Key Laboratory of Materials Synthesis and Processing, Wuhan University of Technology, Wuhan 430070, China

**Keywords:** hygrothermal condition, carbon epoxy, moisture, impact damage

## Abstract

The influence of hygrothermal aging on high-velocity impact damage of carbon fiber-reinforced polymer (CFRP) laminates is investigated. Composite laminate specimens were preconditioned in water at 70 °C. The laminates were subsequently impacted by flat-, sphere-, and cone- ended projectiles with velocities of 45, 68, and 86 m/s. The incident and residual velocities were collected during the impact test. The impact-induced damages were measured by ultrasonic C-scan, a digital microscope system, and a scanning electron microscope. The results show that the hygrothermally conditioned laminates offer a higher energy absorption during high-velocity impact. Due to the weakening of the interlaminar properties, the hygrothermally conditioned laminates are more susceptible to delamination failure, and shear-induced debonding dominates. The projected delamination area increases with the increment of impact velocity. The damaged region becomes close to a circular shape after hydrothermal conditioning, and close to a rhomboidal shape for the dry specimens.

## 1. Introduction

For its high specific strength and specific stiffness, fatigue and corrosion resistance, carbon fiber-reinforced polymer (CFRP) laminates are widely used in aeronautical and nautical structures [1,2]. Humid conditions are known to decrease the mechanical properties of epoxy-based composites that are dominated by matrix or interface. As those structures usually suffer from a poor environment, the effect of hydrothermal conditions on their mechanical properties is a critical issue [3]. Hence, in the last decades, materials science researchers have invested their efforts to conduct moisture-impact studies of CFRP laminates.

In early works, the hygrothermal effects on the mechanical properties and the failure behavior of CFRP composites were firstly studied [3,4,5]. It was found by Woldesenbet et al. [6] that the CFRP had a 12%~14% decrease in quasi-static strength and a 25%~40% increase in dynamic strength after hygrothermal conditioning. Effects on the static mechanical strength of epoxy-matrix unidirectional CFRP composite were studied by Tual et al. [7], in which a decrease of 20%~40% was reported in the failure strengths. Meng et al. [8] found that hygrothermal stress could be induced into the interlaminar shear stress of the CFRP, and reported the induced stress was up to 20% of the interlaminar shear strength. Liu et al. [9] investigated the hygrothermal effects on the mechanical properties of the double lap shear joints of carbon/epoxy composite laminates, which exhibited a decrease of about 24% and 27% in the elastic modulus and tensile strength after exposure to a humid environment at 90 and 95 °C.

Currently, the hygrothermal aging studies of the CFRP composites have focused on the hygrothermal (including water immersion) effect on the resistance of low-velocity impact. In order to study the compression-after-impact (CAI) property of the CFRP laminates after hygrothermal conditioning, Aoki et al. [10] designed drop-weight impact tests with impact velocities ranging from 2.38 to 2.43 m/s. They found that the delamination area and the number of transverse cracks of the wet specimen were smaller than those of the dry specimens. Hosur et al. [11] studied the performance of woven carbon/epoxy laminate after low-velocity impact with energy levels of 15, 30, and 45 J. They concluded that the samples with cold-moist conditioning became plasticized, exhibiting more ductility, and could withstand higher peak loads than the dry ones. Zhong et al. [12,13,14,15] conducted several low-velocity impact experiments on the CFRP laminates. They concluded that the absorbed moisture softened the matrix resin and caused a decrease in the strength and modulus of the laminates. The hygrothermally induced damage postponed the occurrence of fiber debonding, breakage, and multiple matrix cracking, though they declined the impact resistance of the laminates. Eventually, the wet/hot environment improved the impact response of conditioned CFRP laminates. It can be seen from the above studies that the hygrothermal effect on the low-velocity impact resistance of the CFRP laminates has a promoting effect and weakens the damage.

However, the performance of a structure subjected to a low-velocity impact differs with respect to the high-velocity case, so that the high-velocity impact should be conducted further. Unfortunately, there are limited studies focusing on this issue [16,17]. Inspired by this, the influence of hygrothermal condition on the high-velocity impact damage of CFRP plates was studied experimentally [17]. Firstly, the CFRP laminates were conditioned in hot water at 70 °C. Then, three different projectiles were used to impact the CFRP laminates at 45, 68, and 86 m/s. The incident and residual velocities were measured, and the energy absorption performance was analyzed. In addition, an ultrasonic C-scan technique was employed to detect the macro-delamination damage of laminates. A three dimensional-super depth digital microscope (3D-SDDM) and scanning electron microscopy (SEM) were further employed to observe the microscopic features of the sample damage. The influence of the hygrothermal environment on failure performance of the CFRP laminate under high-velocity impact was discussed, both in the macro and micro scales.

## 2. Materials and Methods

### 2.1. Tested Materials

Laminates were fabricated by unidirectional carbon/epoxy prepreg T300/EM112 (manufactured by Jiangsu Hengshen Inc., Danyang, China) that had a resin content of 66% by weight and a cured layer thickness of 0.137 mm. The prepregs were cut into sizes of 115 × 115 mm^2^ and stacked with a sequence of [0/90]_8_. The prepreg laminates were cured in an autoclave (RYG-21, Lontek Inc, Xi’an, China) with the curing cycle plotted in Figure 1. A two holding stages curve was used, where the dwelling stages were 80 °C for 30 min and 130 °C for 120 min, the heating rate was 2 °C/min and the cooling rate was −2 °C/min. The pressure of 0.3 MPa was applied during curing. 

### 2.2. Hygrothermal Conditioning

Prior to subsequent hygrothermal conditioning or impact testing, the cured specimens were conditioned in a dry and vacuum oven (DHG-9626A, JingHong Inc., Shanghai, China) at 70 °C for 24 h, so as to prevent previous moisture absorption. Subsequently, the vacuum oven-dried specimens were immersed in water. A water bath of 70 °C was employed in the immersion to accelerate the aging process, since temperature does not change the saturated moisture content but accelerates the process diffusion [18]. An analytical balancer (AL104, Mettler Toledo, Columbus, OH, USA) with an accuracy of 0.1 mg was used to measure the weight of specimens periodically during the conditioning. The initial mass was recorded as $m_{d}$ before water immersion and $m_{w}$ after that. The moisture absorption of the CFRP laminates sample can be expressed as (1)$$M_{i}=\frac{m_{w}-m_{d}}{m_{d}}\times 100\%$$
where $M_{i}$ represents the moisture absorption of the sample, unit: %; the unit for $m_{d}$ or $m_{w}$ is g.

The duration of the hygrothermal condition was determined by the moisture absorption curve which is discussed later.

### 2.3. Impact Test

Hygrothermal conditioned samples were subjected to high-velocity impact using a 14.5 mm diameter, 1.2 m long light-gas gun loading system. As shown in Figure 2, the impact testing device consists of a tank nitrogen gas supplier, a pressure gauge, a sample holder, two projectile velocity measuring units, and a ballistic paste. The samples were clamped on the holder. Three different projectiles with a diameter of 14.3 mm were used to investigate the effect of the geometry of impactor, including flat-, sphere-, and cone-shaped ends, as shown in Table 1. The incident velocity of the projectile was calibrated for nitrogen gas at various gas pressures. The residual velocity was measured by VISAR (Velocity Interferometer System for Any Reflector) technique. In this study, the saturated samples were subjected to incident velocities of 45, 68, and 86 m/s, respectively. Each test was repeated about 8–12 times. 

### 2.4. Damage Detection

Damaged features of the impacted samples were characterized using the ultrasonic C-scan, 3D-SMMD, and SEM techniques. The damaged laminates were tested by the C-scanning system (UPK-T48-HS, PAC Inc., McLean, VA, USA) with a 10 Hz flat transducer. Additionally, the delamination was post-processed on UTwin^TM^ software (version E3.62). The projected damaged areas from the front to back surfaces of the impacted laminates were then quantified. The damaged laminates were cut by a diamond saw to determine both the damage extent and morphology, with the aid of 3D-SDDM (VHX-600E, KEYENCE Inc., Osaka, Japan) and SEM (SH-1500, HIROX Inc., Tokyo, Japan) to visualize the thorough failure on the cross section. 

## 3. Results and Discussion

### 3.1. Moisture Absorption Behavior

The moisture absorption behavior is shown in Figure 3, where the percent weight gain is plotted as a function of the immersion time (s). The weight gain value for such a case at 600 h (25 days) was 1.748 ± 0.04%. Afterwards, the moisture absorption goes to a stable stage. Since no significant increase of moisture content was observed after 40 days, the samples at 40 days were used as saturated ones in this study. The average value of saturated moisture absorption was 1.775%. Using Fick’s law [19], the diffusion rate was 6.182 × 10^−7^ mm^2^/s.

Figure 4 shows the microscopic images (magnified 500 times) of the specimen under a dry condition and a wet condition at 70 °C for 30 days using the 3D-SDDM technique. It shows that no damages (i.e., matrix cracking, interlayer splitting, and fiber disconnection) were observed, which indicates that the hygrothermal conditioning at 70 °C for 30 days had no significant influence on the sample surface finish and caused no irreversible material structural damage.

### 3.2. Energy Absorption

In this section, the hygrothermal effect on the energy absorption of composites under high-velocity impact conditions has been investigated. To simplify the analysis, the impactor was assumed as a rigid body when the projectile impacted the CFRP laminates, and the energy loss induced by friction was neglected. The loss of kinetic energy was regarded to be completely absorbed by the laminates. 

The incident kinetic energy of the projectile $E_{s}$ is
(2)$$E_{s}=\frac{1}{2}mv_{s}^{2}$$

Additionally, the residual kinetic energy $E_{r}$ is
(3)$$E_{r}=\frac{1}{2}mv_{r}^{2}$$
where *m* is the mass of the projectile, $v_{s}$ represents the initial impact velocity, and $v_{r}$ represents the residual velocity after perforation.

Using Equations (2) and (3), the energy absorption E would be expressed as
(4)$$E=E_{s}-E_{r}=\frac{1}{2}m\left(v_{s}^{2}-v_{r}^{2}\right)$$

Figure 5 presents the energy absorbing performances of all the laminates. The impact velocities were not exactly equal with the presented 45, 68, and 86 m/s, respectively, because the projectile mass was different while the nitrogen gas pressure was fixed for three pressure levels. Because of the ricochet of the projectiles, the residual velocity was not collected effectively, and data points are calculated inadequately in several cases. For example, with the impact velocity of 45 m/s, it was found in the experiments that the flat- and sphere- ended projectile did not perforate the laminates and ricocheted. Particularly for the flat end, the ricochet was sensitive to the non-uniform reaction on the contact surface [20]. The corresponding ricochet route could be irregularly oblique. Along with the small velocity, it was difficult to accurately and effectively capture the ricochet path. Then the time intervals in those cases could not be recorded effectively. Thus, the limited corresponding residual velocities, which were inadequately measured, were only qualitatively given for the trending analysis.

For other impact cases with velocities of 68 and 86 m/s, the projectiles perforated the CFRP laminates. With the incipient and residual velocities, the corresponding kinetic energy loss was calculated using Equations (2)–(4). Compared with the dry specimens, the saturated specimens have higher kinetic energy loss, indicating that more energy was absorbed during the impact incidents of the wet ones. When the dry specimen is impacted by the cone-ended projectile at a velocity of 86 m/s, the energy loss is 13.5% greater than that for the dry one; when the projectile is a flat-ended projectile, the energy loss for the wet specimen is 4.0% greater than that for the dry specimen. In some cases, evident standard deviations could be found. This is owing to a few scattered data in the single case, which is supposedly induced by the difficulty in controlling the incident angle, which may lead to variable penetration angles and positions, as well as after-penetration paths. As the trend of the kinetic energy loss vs. impact velocity remains in the vast majority of cases, the law that energy absorptions are higher in the wet specimens should be appropriate for the cases of the presented impact velocities. 

Moreover, this law could be appropriate for the presented different projectile ends. Of the cases with the same incident velocity and environmental condition, the loss of kinetic energy for the flat-ended projectile was the highest, followed by the sphere-ended one, and the cone-ended one was the lowest. For instance, when the wet specimen was impacted by the flat-ended projectile at a velocity of 68 m/s, the energy loss was 10.8% greater than that of the specimens impacted by the sphere-ended one, and 20.2% greater than that of the case with the cone-ended projectile. It indicates that the saturated specimens have the strongest resistance to the incident impact of the flat-ended projectile, while the weakest resistance refers to the cone-ended projectile. 

### 3.3. Impact Damage Characteristics

#### 3.3.1. Morphological Features

Morphological study of the surface damage for both the dry and saturated specimens shows that the visualized geometries of the rupture are different when projectile ends are different, indicating different types of perforation. 

As seen in Figure 6, an indentation left in the front face of the dry laminates after the impact of the sphere-ended projectile with the velocity of 45 m/s, followed by a bulge in the back face. In the saturated specimens, due to the hygrothermal effect, the indentation was centralized in a smaller region, along with a more evident rupture. In the front faces, the collapse was also deeper than the dry one, and several transverse cracks occurred near the indentation. Correspondingly, the bulge was progressive in the back face, and the edge of some fibers appeared tilted, which suggests that fiber and matrix slipped relative to each other. At the other end of these fibers, strips of transverse cracks occurred, and it further indicates the weak bonding between fiber and matrix.

Figure 7 presents the morphology of the laminates after the impact of the flat-ended projectile with the velocity of 68 m/s. Although the impact velocity was higher, the projectile bounced back, and no apparent indentations were visualized in the front face after the flat-ended projectile impacted the dry specimen. Only a bulge of length 10.52 mm and width 33.3~41.22 mm could be observed in the back face. For the saturated specimen, an indentation in the front face was left after impact. Near the indentation edge, two transverse cracks of lengths 60.64 and 26.00 mm appeared. The bulge formed in the back face occurred tilted, with a length of 60.64 mm, which was consistent with the indication that the weak bonding between fiber and matrix exists in the wet specimens.

The morphology of the laminate faces after the impact of the cone projectile at 86 m/s is presented in Figure 8. Due to the perforation, a hole was left in the front faces. A combination of the interlaminar separation (delamination) and tensile crack propagation was followed in the back faces, along with cruciform-type bulges. In the dry samples, the free edge of the damaged front layer still overlapped on the fracture edge after the perforation. Near the hole, several extended cracks accompanied. In the back face, a 44.81 mm length of transverse crack occurred, along with many separated fibers tilted. Instead, for the wet sample, the fibers near the fracture were clearly broken and disconnected from the neighboring panel. The fracture morphology was shaped like a “window”. The transverse fibers upon the back face were also broken, while the fracture below the last layer can be visualized.

The above morphological difference between the dry and saturated samples was also consistent with the saturated samples absorbing more energy, so that more progressive fracture morphologies were caused. Furthermore, it also suggests the fiber/matrix interface of the saturated samples had a lower capability to resist the perforation, during which fibers were broken and the damaged surface was rougher.

#### 3.3.2. Measurement of Delamination Area

The projected areas of internal delamination were measured using ultrasonic C-scan. Combined with the rupture images, the hole produced by a higher velocity impact has the shape of the intersection between the laminate and the trajectory of the projectile. Thus, the C-scan result of the delamination was localized around this orifice. In Figure 9, Figure 10 and Figure 11, the darkest region in the center of the laminate represents the projected damaged region, around which the dark grey represents the fiber peeling fractures.

Figure 9 plots the C-scan characteristic of the specimens after the impact of the flat-ended projectile at 45, 68, and 86 m/s in Figure 9a–c “dry” and Figure 9d–f “wet”. In such cases, the damaged areas in the saturated specimens in Figure 9d–f were close to the ellipse shape, whereas those in the dry one in Figure 9a–c were approximately rhomboidal in shape. Such differences of the damaged areas between the dry samples and the saturated sample could be also seen in other projectile ends, as shown in Figure 10 and Figure 11.

To quantitatively understand the delamination area, the damaged areas for all those cases are presented in Figure 12. The damage geometry as observed through the C-scan images yielded higher values than those calculated from visual observation. Along with Figure 9, Figure 10 and Figure 11, this figure displays that the damage extent increases from the dry samples to the saturated samples, when the impact velocities or the projectiles are different. Under the impact of the same velocity of 68 m/s with the sphere-ended projectile, the wet specimen has a 17.8% increase in the damaged area relative to the dry specimen; the damaged area of the wet specimen increases 18.07% when the projectile is cone-shaped. These results indicate that the delamination damage is more likely to occur due to the severe degradation of the interlayer performance when the specimen is hygrothermally conditioned. However, this law is found to be somewhat different from that of the energy absorption in Figure 5, which further indicates that the kinetic energy dissipates in other ways other than the occurrence of delamination, such as fiber fracture and the matrix cracking. The results also show this increase grows limitedly when the impact velocity becomes high to a certain level or the projectile end becomes sharp to a certain extent. For example, the wet specimen impacted at a velocity of 68 m/s with the cone-ended projectile had a 5.0% increase in the damaged area, compared to the dry specimen. It could be found that the wet specimens impacted by the velocities of 68 and 86 m/s with the flat ended projectile had a relatively higher damaged area than others, which might be related with the different impact responses of the wet laminates by the flat-ended projectile. However, the reason is still unclear. 

According to the principle of energy conservation, the external work on the laminates is almost transformed to the work of the internal stress [21], namely, the energy absorption of the laminates. When the delamination area becomes larger, it indicates that more regions of the laminates participate in the work or more energy is absorbed. When more progressive damages and larger delamination areas in the saturated specimen were observed, it had a better capability of energy absorption [22]. Furthermore, the damage extent or delamination areas for the dry sample and the saturated sample have small discrepancies when subjected to high-velocity perforation, which explains the limited improvement in energy absorption.

#### 3.3.3. Microscopic Damage Characteristics

The cross-sectional features of the fiber/matrix interface using the 3D-SDDM technique (10 μm~1 mm) are presented in Figure 13 and Figure 14 where the impact case of the cone projectile at 45 m/s is given. The hygrothermal effect can be presented by comparing the micro images of the dry specimen, as shown in Figure 13, with those of the wet specimen, as shown in Figure 14.

Comparing Figure 13 with Figure 14 shows that the rupture development in the dry specimen includes matrix cracking, interface debonding, and fiber breaking through the cross section. Micro-cracks in the matrix were produced due to the impact. Part of the matrix near the indentation was broken into particles and scattered between the fibers. The fiber/matrix debonding and delamination were clearly visualized. Relatively, the saturated specimen mainly exhibited evident deflection in the back face, indicating the easiness for the cone projectile to wedge through the composite face. Thus, the perforation should be featured with more fiber stretching, higher tensile and flexural modulus, fiber breakage, and consequently higher energy absorption. It indicates the composite should have higher resistance to fracture or impact damage.

To further understand the damage mechanism, Figure 15 and Figure 16 present the SEM characteristics of the laminates at the core fracture regions of the dry and saturated samples. They particularly provide the damage features of the fiber fracture, the matrix crack, and the interlaminar fracture.

In Figure 15, the compressive, as shown in Figure 15a, and shear damages, as shown in Figure 15b, in the dry specimens were evidently found on the front laminate layer. It indicates that the fiber/matrix cracking in the front face was caused by compression and shear action. Subsequently, the broken fibers in the core fracture were seen still connected, and the extensive debonding and delamination could be observed along the fiber, as shown in Figure 15d. It could be explained by that the fibers and matrix moved with the projectile, while still bonding to the panel and retaining the resistance to the impact. The effective mass of the impactor increased due to the addition of the shear plugging [23] so that the penetration velocity decreased. The friction force or the shear stress imposed on the laminates also decreased. Moreover, crack propagation was transmitted to the neighboring undamaged matrix and fibers. The damaged features, as shown in Figure 15a, indicates that the fiber/matrix cracking in the back face is caused by stretching and delamination. As the damage in the front layer is more serious with spallation state, the compression and shear action in the front face was dominant in the damage process for the dry samples.

As plotted in Figure 16, the matrix near the fracture was crashed into pieces and almost shed from the fiber, as shown in Figure 16b. Most of the matrix in the back faces slid and only a small part was still attached to the fiber, as shown in Figure 16c. The fiber fracture appeared as a blunt shape without fiber connection. Combined with the perforation and the delamination area, the prevailing failure modes of the impact damage in the saturated specimens are a shear-induced delamination failure of the fiber/matrix interface and tensile failure of the fibers. However, when the velocity was much higher, it could be known that the delamination failure might have completed before some fibers stretching could respond, as shown in Figure 8c. Without the responses of the matrix cracking and fragmentation to resist damage initiation, the fibers were solely broken due to the imposed action exceeding their strength limits. Thus, the shear-induced delamination was the main failure mode in the high-velocity impact in wet samples, as the tension of the composite plies could provide auxiliary resistance to damage. 

Combined with the higher energy absorption in the saturated specimen, it explains that the hygrothermal conditioning increases the resin ductility and offered the laminate a higher material resistance to damage initiation. When subjected to high-velocity impact, the damage geometry in perforated CFRP targets can be interpreted by analogy with the plastic behavior of ductile metallic targets. Furthermore, it explains the relatively larger delamination area with an approximately ellipse shape in the saturated samples.

## 4. Conclusions

This study successfully investigated the performance of hygrothermally conditioned CFRP laminates under high-velocity impact. The impact damage was characterized by ultrasonic C-scan, SEM, and 3D-SDDM techniques. The following conclusions can be drawn from the results.
After the hygrothermal conditioning, the interfacial performance of the CFRP laminates decreased.The saturated samples have better energy absorption. The hygrothermal conditions also offer higher resistance to damage initiation due to the enhancement of the ductility in the resin.With the micro-imaging of the fracture, the shear-induced delamination and the debonding of the composite plies were observed, along with some tensile fractures of fibers. Those patterns suggest that the failure mode in the saturated samples was a shear-induced delamination.With the increase in impact velocity, the energy absorption and the projected damage area in the CFRP laminates increases. Once the velocity increases to a certain level, the growth rate of the damaged area was limited in the hygrothermally conditioned ones.The energy absorption was also affected by the projectile geometry for both the “dry” and “wet” samples. Under the presented velocity range, *E*(Flat) > *E*(Sphere) > *E*(Cone) when the projectile could perforate the laminate.Due to the alteration of the failure mechanism, the damaged regions of the laminates became elliptically shaped after hygrothermal conditioning, and rhomboidally shaped for the unconditioned specimens.

## Figures and Tables

**Figure 1 materials-11-02525-f001:**
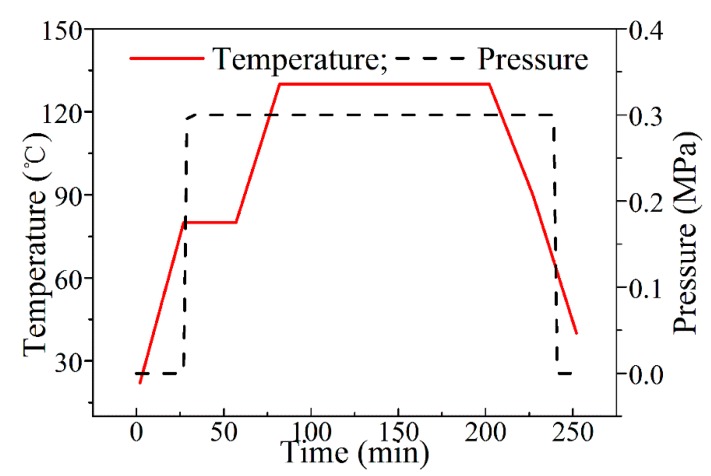
Curing cycle adopted for curing of carbon fiber-reinforced polymer (CFRP) laminates in an autoclave.

**Figure 2 materials-11-02525-f002:**
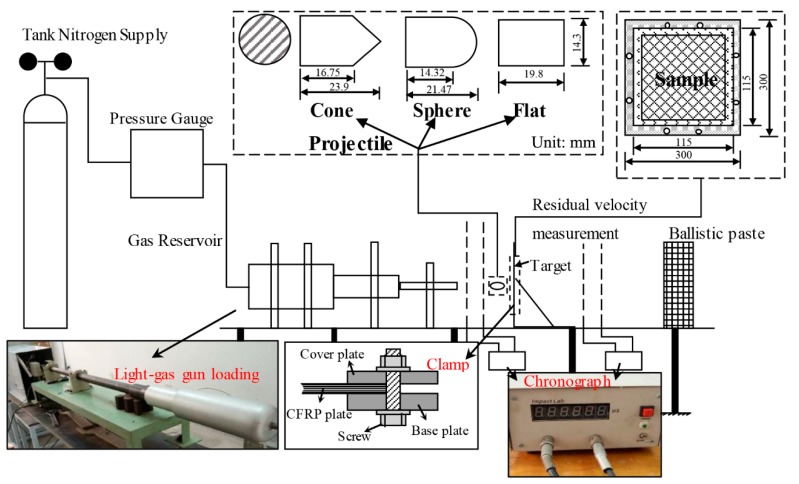
Schematic illustration of high speed impacting system.

**Figure 3 materials-11-02525-f003:**
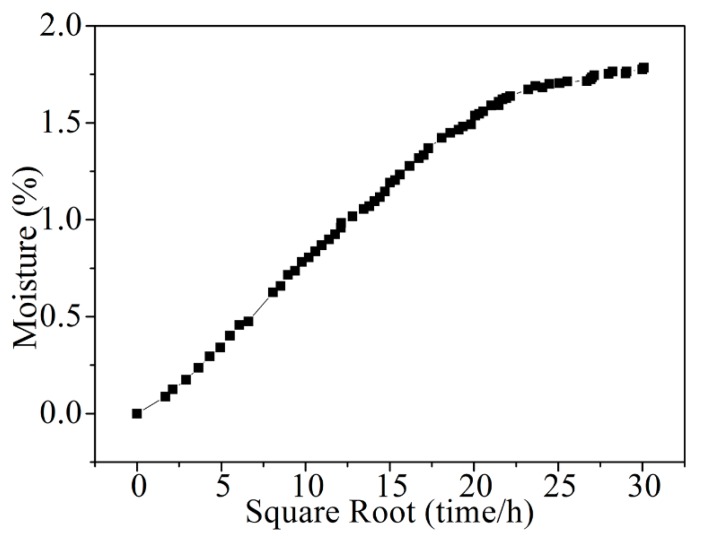
Moisture uptake profile of the CFRP laminates exposed to water at 70 °C.

**Figure 4 materials-11-02525-f004:**
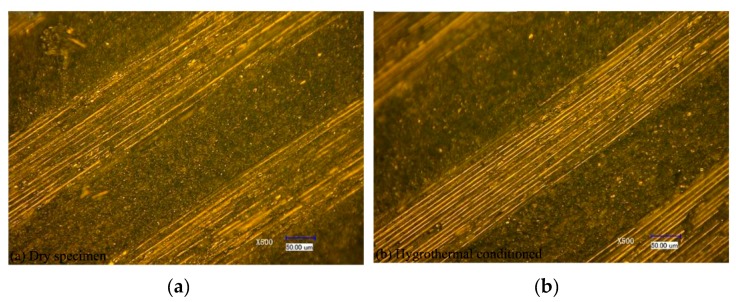
Images of the CFRP laminates in “dry” (**a**) and “wet” (**b**), hygrothermal (70 °C) conditions for 720 h using the three dimensional-super depth digital microscope (3D-SDDM) technique.

**Figure 5 materials-11-02525-f005:**
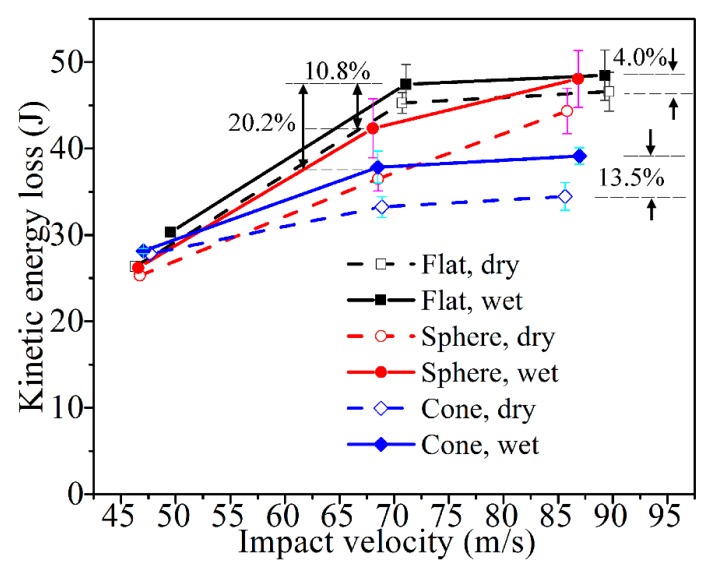
Incident impact energy loss of the CFRP laminates in “dry” and “wet”.

**Figure 6 materials-11-02525-f006:**
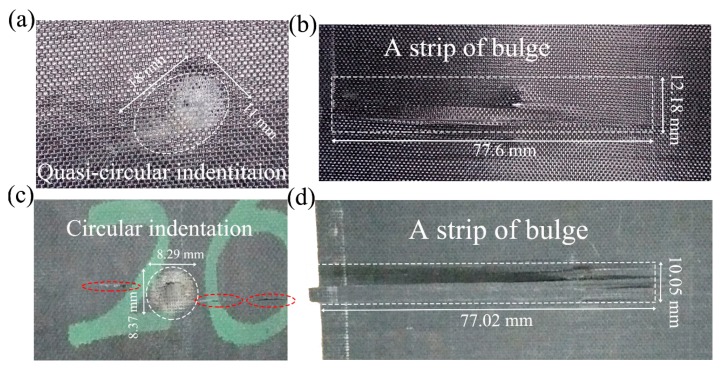
Photos of (**a**) the front and (**b**) back faces of a dry specimen, (**c**) the front and (**d**) back faces of a saturated specimen under an impact velocity of 45 m/s with the sphere-ended projectile.

**Figure 7 materials-11-02525-f007:**
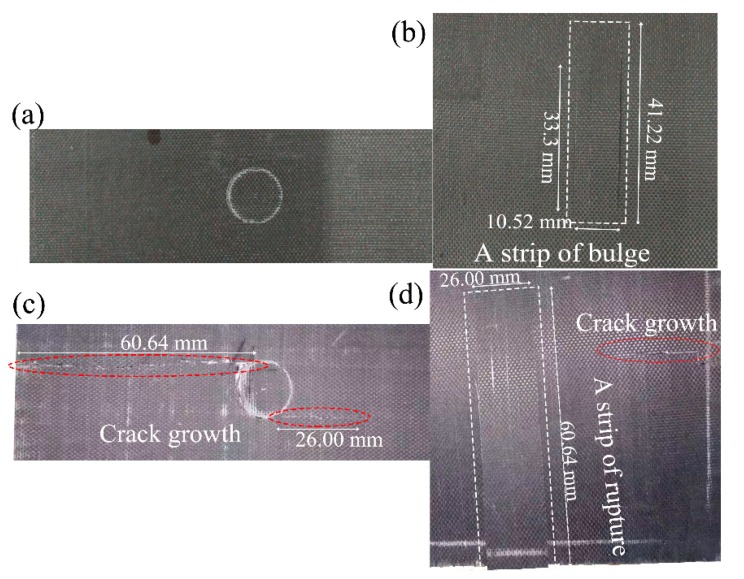
Photos of (**a**) the front and (**b**) back faces of a dry specimen, (**c**) the front and (**d**) back faces of a saturated specimen under an impact velocity of 68 m/s with the flat-ended projectile.

**Figure 8 materials-11-02525-f008:**
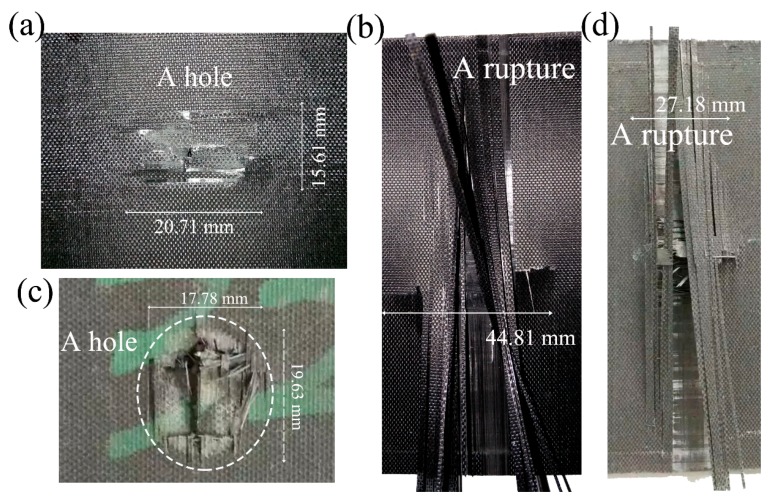
Photos of (**a**) the front and (**b**) back faces of a dry specimen, (**c**) the front and (**d**) back faces of a saturated specimen under an impact velocity of 86 m/s with the cone-ended projectile.

**Figure 9 materials-11-02525-f009:**
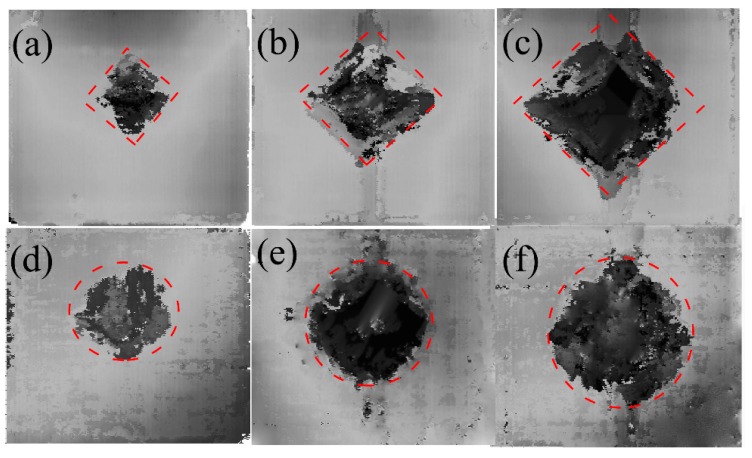
C-scan characteristic of the specimen after the impact velocities of (**a**) 45 m/s, (**b**) 68 m/s, and (**c**) 86 m/s in “dry” and of (**d**) 45 m/s, (**e**) 68 m/s, and (**f**) 86 m/s in “wet” for the flat-ended projectile.

**Figure 10 materials-11-02525-f010:**
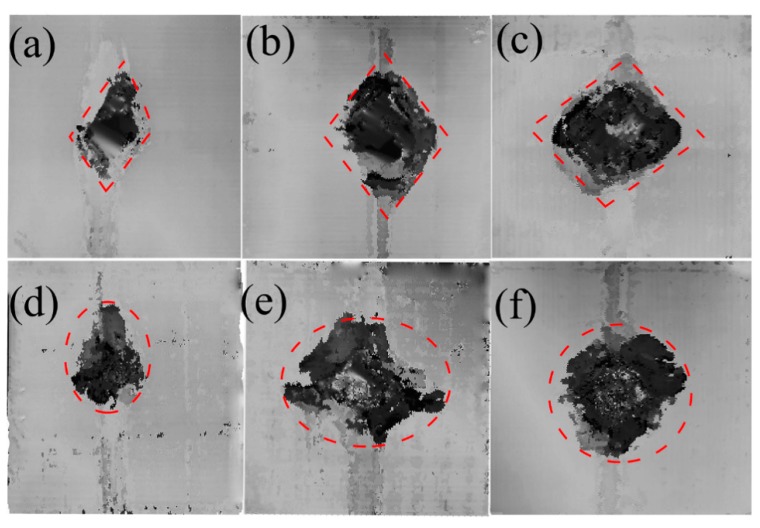
C-scan characteristic of the specimen after the impact velocities of (**a**) 45 m/s, (**b**) 68 m/s, and (**c**) 86 m/s in “dry” and of (**d**) 45 m/s, (**e**) 68 m/s, and (**f**) 86 m/s in “wet” for the sphere-ended projectile.

**Figure 11 materials-11-02525-f011:**
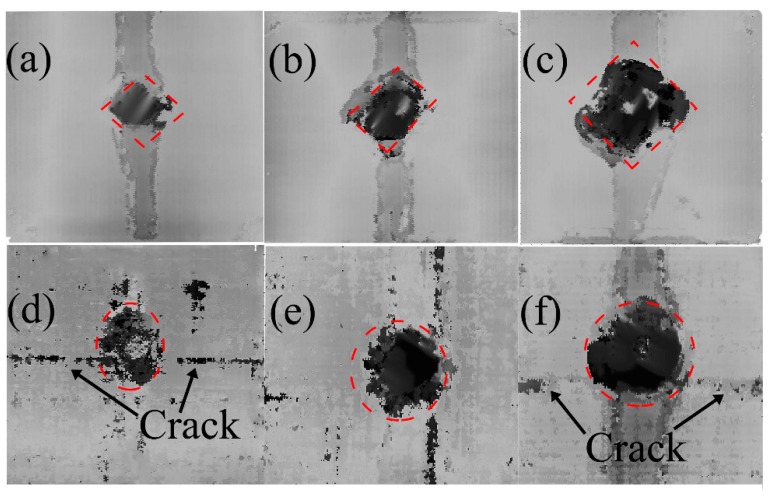
C-scan characteristic of the specimen after the impact velocities of (**a**) 45 m/s, (**b**) 68 m/s, and (**c**) 86 m/s in “dry” and of (**d**) 45 m/s, (**e**) 68 m/s, and (**f**) 86 m/s in “wet” for the cone-ended projectile.

**Figure 12 materials-11-02525-f012:**
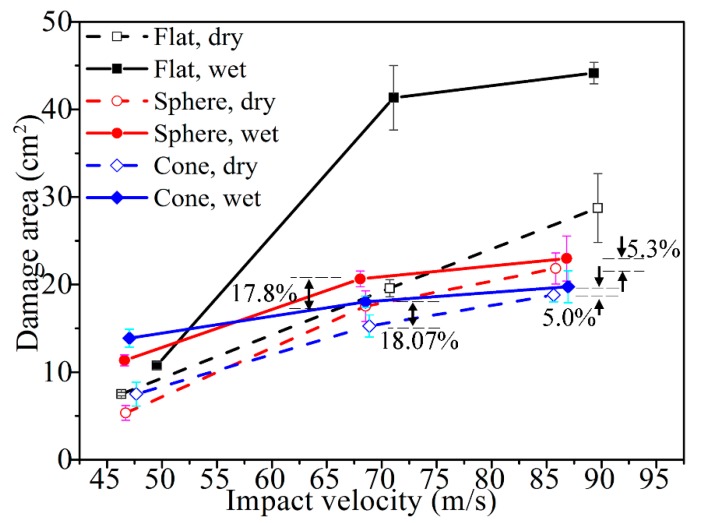
Projected damaged areas of the CFRP laminates in “dry” and “wet”.

**Figure 13 materials-11-02525-f013:**
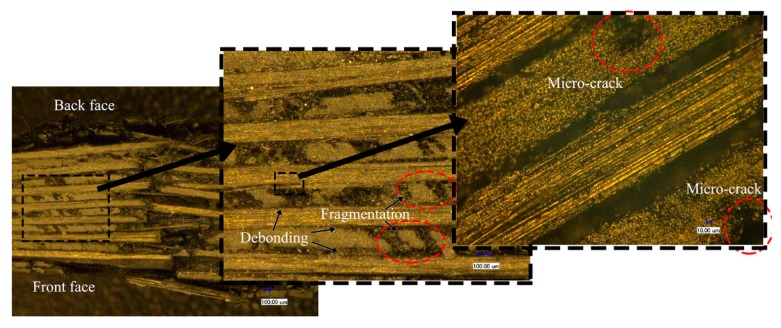
Impact-damaged images of the dry specimen subjected to the impact velocity of 45 m/s using the 3D-SDDM technique.

**Figure 14 materials-11-02525-f014:**
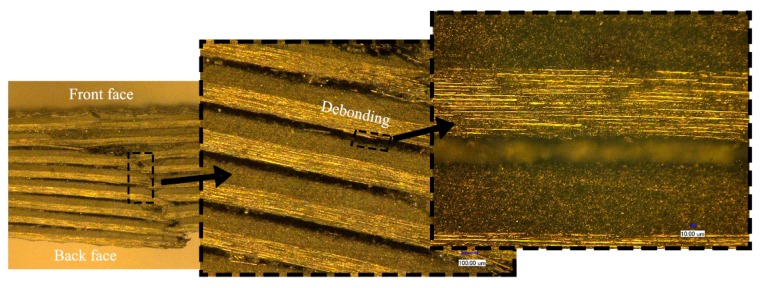
Impact-damaged images of the saturated specimen subjected to the impact velocity of 45 m/s using the 3D-SDDM technique.

**Figure 15 materials-11-02525-f015:**
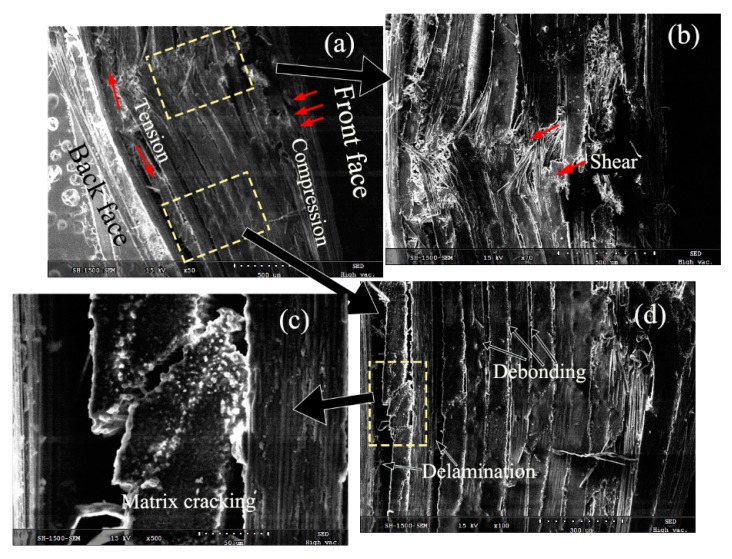
Fracture patterns of the dry specimen subjected to the impact velocity of 45 m/s using the SEM technique. (**a**) Compressive damage; (**b**) Shear damage; (**c**) Matrix cracking; (**d**) Extensive debonding and delamination.

**Figure 16 materials-11-02525-f016:**
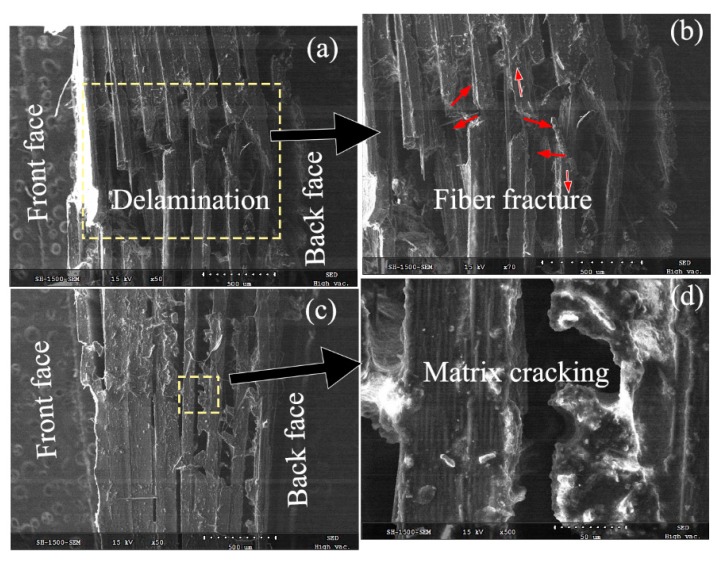
Fracture patterns of the saturated specimen subjected to the impact velocity of 45 m/s using the SEM technique. (**a**) Delamination damage; (**b**) Fiber facture; (**c**) Matrix/fiber interface; (**d**) Matrix cracking.

**Table 1 materials-11-02525-t001:** Different conditions in the impact test (with water immersion at 70 °C).

Case	Impact Velocity (m/s)	Projectile End Shape	Projectile Mass (g)
S2-T-V1	45	Flat	24.21
S2-T-V2	68	Flat	24.21
S2-T-V3	88	Flat	24.21
S3-T-V1	45	Sphere	24.14
S3-T-V2	68	Sphere	24.14
S3-T-V3	88	Sphere	24.14
S1-T-V1	45	Cone	24.32
S1-T-V2	68	Cone	24.32
S1-T-V3	88	Cone	24.32

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
