# Peer review of "Damage of Hygrothermally Conditioned Carbon Epoxy Composites under High-Velocity Impact"

_materials, 2018, doi:10.3390/ma11122525_

Round 1
Reviewer 1 Report
The submitted article (materials-396769-v1) entitled: “Damage of hygrothermally conditioned carbon epoxy composites under high-velocity impact” contains original contribution to the study of hygrothermally conditioned Carbon Fibre Reinforced Polymer (CFRP) laminates under high-velocity impact. CFRP laminate damages were caused by flat-, sphere- and cone-shaped projectile impact tests and afterwards they were characterized by ultrasonic C-scan, digital microscope system and scanning electron microscope techniques. The paper includes a well-documented and interesting experimental program. This study deals with a subject that is still open to question. The manuscript is neatly written, well-structured and easily understood. Figures are also precise and helpful. For all the aforementioned reasons, I believe that this research is definitely worthy of publishing. Some minor revisions based on the below mentioned suggestions will make the authors to feel happier for their good work:
- The citation of the references in the manuscript should be in the same sentence that is referred (before full-stop) as for example in line 30: “…nautical structures [1][2]. Humid condition…” instead of “…nautical structures.[1][2] Humid condition…”.
- The symbols used in equations (2)-(4) should properly be explained in the manuscript.
- More discussion of the results derived from Figs. 5 and 12 would be very helpful for the readers of the paper.
Author Response
Reply to Questions of Reviewer #1
General comments and Suggestions for Authors
The submitted article (materials-396769-v1) entitled: “Damage of hygrothermally conditioned carbon epoxy composites under high-velocity impact” contains original contribution to the study of hygrothermally conditioned Carbon Fibre Reinforced Polymer (CFRP) laminates under high-velocity impact. CFRP laminate damages were caused by flat-, sphere- and cone-shaped projectile impact tests and afterwards they were characterized by ultrasonic C-scan, digital microscope system and scanning electron microscope techniques. The paper includes a well-documented and interesting experimental program. This study deals with a subject that is still open to question. The manuscript is neatly written, well-structured and easily understood. Figures are also precise and helpful. For all the aforementioned reasons, I believe that this research is definitely worthy of publishing. Some minor revisions based on the below mentioned suggestions will make the authors to feel happier for their good work:
A: Firstly, we would like to thank the referee for spending much time reviewing our manuscript to materials. Based on the reviewer’ useful comments, we have seriously revised the paper. In this response letter, we have provided a written reply (below) to all the questions raised by the referee. If the referee has further questions, please let us know. We shall further work on them.
Q1- The citation of the references in the manuscript should be in the same sentence that is referred (before full-stop) as for example in line 30: “…nautical structures [1][2]. Humid condition…” instead of “…nautical structures.[1][2] Humid condition…”.
Re: We would like to thank the referee for carefully reading this manuscript and pointing out this. We have corrected it in the revised manuscript. Please refer to line 30, line 33, line 36, line 64, line 277.
Q2- The symbols used in equations (2)-(4) should properly be explained in the manuscript.
Re: Much thanks to the referee for the kind comments. According to the comments, we have supplemented the explanations of symbols, such as energy and mass of the projectile. Please refer to lines 145 ~ line 146.
Q3- More discussion of the results derived from Figs. 5 and 12 would be very helpful for the readers of the paper.
Re: Thanks very much for the referee providing the suggestions. According to the suggestion, we have added the content of discussion for the results in Figs 5 and 12. Please refer to lines 151-178 and lines 255-269 in the revised paper.
Reviewer 2 Report
Dear Authors,
First of all, I wish to congratulate the authors because their work is very interesting from experimental point of view of the damage of hygro-thermally conditioned carbon epoxy composites under high-velocity impact.
The research topic of the manuscript fits on the journal purpose. The conclusions are adequately supported by the data presented.
The materials tested are carbon fiber reinforced polymer (CFRP) laminates that are widely used in aeronautical and nautical industries. The topic about damage of these kinds of composites under high-velocity impact, is sufficiently interesting and novel in order I recommend the publication of this paper after some minor improvements describes bellow.
In order to improve this interesting manuscript, I recommend the following:
Section “2.1. Sample and preparation”
-I recommend to rename this section as “Tested materials”.
-The size of the composite sample is 115x115 mm2 in accordance with Fig. 2. Because “The pregregs were cut into sizes of 115 × 115 mm2 and stacked with 77 a sequence of [0/90]8”, then the following text should be deleted: “The laminates were cut into samples after curing with diamond plate saw.”
-Number 8 should be formatted as subscript in the text “[0/90]8”.
-The pressure “300 kPa” should be expressed in MPa (i.e. 0.3MPa). I recommend also to express the pressure in MPa in the graph shown in the Fig.1.
Section “2.2. Hygrothermal conditioning”
-Insert a new reference in order to justify the text “… since temperature does not change the saturated moisture content but accelerate the process diffusion.”
-The letter “G” usually denotes the weight. So, I recommend to denote by “m” the mass instead of “G”. You can use “w” as subscript in case of the mass of the wet sample and “d” or “0”- subscript for dried sample.
Section “2.3.Impact test”
-How many samples are tested in case of each velocity and each kind of projectile shape shown in Table 1?
-Is there a standard used to achieve the high-velocity impact test?
Section “3.1 Moisture absorption behavior”
-I recommend to draw the absorbed moisture (%) related with the square root of time (expressed in the square root of hours) in Fig. 3 in order to demonstrate that this curve obeys Fick’s law in case of the composite material involved in this study in accordance with the literature concerning to the moisture absorption in fiber reinforced plastics. I would like also to compute the diffusion coefficient denoted with D by using Fick’s law and add this result in the manuscript.
Section “3.2. Energy absorption”
-The notations for the initial impact velocity and residual velocity should be written immediately after relation (2) or (3), respectively.
-Line 145: The text “Due to rebound of the projectiles…” should be replaced with “Because of the ricochet of the projectiles…”.
-The authors should explain that the in Fig. 5, the impact velocities are not equal exactly with 45m/s, 68m/s and 86m/s respectively, because the projectile mass is different while the helium gas pressure was fixed for three pressure levels.
-Regarding Fig. 5, the authors could add some remarks regarding how many times is greater the absorbed energy in case of the flat projectile than the corresponding value recorded in case of the cone or sphere projectile in case of wet / dried samples for each impact velocity.
-Fig. 5 shows a large scattering of the results concerning to the kinetic energy loss recorded in case of a certain type of projectile or a certain impact velocity. How you can justify? How many samples were tested in each case? The adding of the values of the standard deviation should be analysed.
Section “3.3.1 Morphological feature”
-The labels (a), (b), (c), (d) should be written outside of the photos in Figs. 6 and 8.
-The quality of the photos in Figs. 8a and 8b need to be improved because these are not very clear.
Section “4. Conclusion”
The conclusions are very good supported by the results shown in the manuscript.
Author Response
Reply to Questions of Reviewer #2
General comments and Suggestions for Authors
First of all, I wish to congratulate the authors because their work is very interesting from experimental point of view of the damage of hygro-thermally conditioned carbon epoxy composites under high-velocity impact.
The research topic of the manuscript fits on the journal purpose. The conclusions are adequately supported by the data presented.
The materials tested are carbon fiber reinforced polymer (CFRP) laminates that are widely used in aeronautical and nautical industries. The topic about damage of these kinds of composites under high-velocity impact, is sufficiently interesting and novel in order I recommend the publication of this paper after some minor improvements describes bellow.
A: Firstly, we would like to thank the referee for spending much time reviewing our manuscript to materials. Based on the reviewer’ useful comments, we have seriously revised the paper. In this response letter, we have provided a written reply (below) to all the questions raised by the referee. If the referee has further questions, please let us know. We shall further work on them.
In order to improve this interesting manuscript, I recommend the following:
Section “2.1. Sample and preparation”
Q-1 I recommend to rename this section as “Tested materials”.
Re: Thanks for the kind suggestion. According to the kind suggestions from the referee, we have rewritten the subtitle of this section as “Tested materials”. Please refer to line 77.
Q-2 The size of the composite sample is 115x115 mm2 in accordance with Fig. 2. Because “The prepregs were cut into sizes of 115 × 115 mm2 and stacked with 77 a sequence of [0/90]8”, then the following text should be deleted: “The laminates were cut into samples after curing with diamond plate saw.”
Re: Much thanks to the referee for pointing out this. We have deleted the mentioned sentence. Please refer to lines 80-81.
Q-3 Number 8 should be formatted as subscript in the text “[0/90]8”.
Re: Thanks very much for pointing out this. We have reformatted the number 8 as a subscript in “[0/90]8”. Please refer to line 81.
Q-4 The pressure “300 kPa” should be expressed in MPa (i.e. 0.3MPa). I recommend also to express the pressure in MPa in the graph shown in the Fig.1.
Re: Thanks for the referee providing this kind suggestion. We have corrected the unit and replaced the unit “kPa” with MPa both in the manuscript and in the figure. Please refer to line 84 and Figure 1.
Section “2.2. Hygrothermal conditioning”
Q-5 Insert a new reference in order to justify the text “… since temperature does not change the saturated moisture content but accelerate the process diffusion.”
Re: Much thanks to the referee for the kind comment. We have added a previously published reference from Shen et al. (Shen & Springer, 1976) (DOI:10.1177/002199837601000101). Please refer to line 93.
Q-6 The letter “G” usually denotes the weight. So, I recommend to denote by “m” the mass instead of “G”. You can use “w” as subscript in case of the mass of the wet sample and “d” or “0”- subscript for dried sample.
Re: Yes, the referee is right and thanks for the kind suggestions. We have replaced “G” with “m”, as well as “i” with “w” and “0” with “d”. Please refer to Equation (1) and lines 96-97.
Section “2.3.Impact test”
Q-7 How many samples are tested in case of each velocity and each kind of projectile shape shown in Table 1?
Re: Thanks for the kind comments. 8 to 12 samples were employed for each case with specified velocity and projectile shape. And the number of effective tests was at least 5 for either the dry specimen or the wet case.
Q-8 Is there a standard used to achieve the high-velocity impact test?
Re:Thanks for the kind comments. Our high-velocity test was designed referring to the ASTM standard (ASTM D5229-2004, 2010)(DOI: 10.1520/D5229), as well as the previous studies of high-velocity impact test (Lee & Sun, 1993) (DOI:10.1016/0266-3538(93)90069-S). According to the work of Lee et al., the incipient velocities were chosen according to the involved velocities from 24 m/s to 96 m/s. The velocity is controlled by a tank nitrogen gas supplier and a pressure gauge.
Section “3.1 Moisture absorption behavior”
Q-9 I recommend to draw the absorbed moisture (%) related with the square root of time (expressed in the square root of hours) in Fig. 3 in order to demonstrate that this curve obeys Fick’s law in case of the composite material involved in this study in accordance with the literature concerning to the moisture absorption in fiber reinforced plastics. I would like also to compute the diffusion coefficient denoted with D by using Fick’s law and add this result in the manuscript.
Re: Thanks for the kind comments and suggestions. We have reformatted Fig. 3 according to the comments with the square root of hours. Also, the diffusion coefficient using Fick’s Law has been computed and given. Please refer to Fig.3 and lines 127-128.
Section “3.2. Energy absorption”
Q-9 The notations for the initial impact velocity and residual velocity should be written immediately after relation (2) or (3), respectively.
Re:Again, we would like to thank the referee for carefully reading this manuscript. According to the comments, we have positioned the mentioned notations immediately after Equations (2)-(3). Please refer to lines 145-146.
Q-10 Line 145: The text “Due to rebound of the projectiles…” should be replaced with “Because of the ricochet of the projectiles…”.
Re: Much thanks for the comments, and we have replaced the mentioned sentence with ”Because of the ricochet of the projectiles”. Please refer to lines 149-150.
Q-11 The authors should explain that the in Fig. 5, the impact velocities are not equal exactly with 45m/s, 68m/s and 86m/s respectively, because the projectile mass is different while the helium gas pressure was fixed for three pressure levels.
Re:Thanks very much for the kind suggestions. According to the comments, we have supplemented the explanation for the impact velocities are not equal exactly with the mentioned 45 m/s, 68 m/s, 86 m/s. Please refer to lines 147-149.
Q-12 Regarding Fig. 5, the authors could add some remarks regarding how many times is greater the absorbed energy in case of the flat projectile than the corresponding value recorded in case of the cone or sphere projectile in case of wet / dried samples for each impact velocity.
Re: Thanks for the kind suggestion. We have marked this in the figure. Please refer to Figure 5 and lines 164-166, lines 176-178.
Q-13 Fig. 5 shows a large scattering of the results concerning to the kinetic energy loss recorded in case of a certain type of projectile or a certain impact velocity. How you can justify? How many samples were tested in each case? The adding of the values of the standard deviation should be analysed.
Re: Thanks for this comments.
Firstly, it should be noticed that the high-velocity impact test still has some uncontrollable factors even though they are almost successfully completed. The first one is the imperfect controlling of the incident angle, which is the dominated factor, since the projectile is very sensitive to the incipient posture (Gupta & Madhu, 1992). Slight error in the incident angle may deviate the after-penetration or ricochet path (Johnson, Sengupta, & Ghosh, 1982), owing to the moment formulated by the contact force and the eccentric distance. Particularly for the blunt-ended projectile, there even exists additional influence from the air resistance. Obviously, an oblique route leads to more energy loss and longer path. The second one should be noted is the variation of the impact velocity. Owing to that the velocity of projectile is controlled by the nitrogen pressure, the impact velocities are not exactly the same for the cases with specified velocity and projectile. The third one is due to the processing variability of the specimen. It would lead to the local discrepancies in the porosity and matrix content which would introduce discrepancy of impact response. The last one is the mechanical error of gas gun system, such as the pressure and the updated posture of the clamping device after each impact. It is right that single factor may have slight influence, so that most of the data are convergent. With the combined influence of those factors, a few scattered data were found in some cases, which showed an evident standard deviation.
Secondly, there are 8-12 samples tested in each case (for each impact velocity and each kind of projectile),and the effective samples for either the dry state or the wet state are at least five.
Thirdly, the standard deviations have been analyzed in the presented paper. Please refer to lines 166-172.
Section “3.3.1 Morphological feature”
Q-14 The labels (a), (b), (c), (d) should be written outside of the photos in Figs. 6 and 8.
Re: Thanks for the referee pointing out this. According to the comments, we have reformatted the labels “a”, ”b”, ”c”, ”d” in Figures 6 and 8, so as to make them clear. Please refer to Figures. 6 and 8.
Q-15 The quality of the photos in Figs. 8a and 8b need to be improved because these are not very clear.
Re: Thanks very much for the kind comments. And we have increased the quality of Fig. 8a and 8b, and the size of Fig. 8. Also, the sizes of Figs. 6 and 7 are increased to keep them in consistence. Please refer to Fig 8, as well as Figs. 6 and 7.
Section “4. Conclusion”
Q-16 The conclusions are very good supported by the results shown in the manuscript.
Re: Thanks very much for the referee giving the kind comments.
Reference
ASTM D5229-2004. (2010). ASTM D 5229– 92 – Standard Test Method for Moisture Absorption Properties and Equilibrium Conditioning of Polymer Matrix Composite Materials. Annual Book of ASTM Standards, 92(Reapproved), 1–13. https://doi.org/10.1520/D5229
Gupta, N. K., & Madhu, V. (1992). Normal and oblique impact of a kinetic energy projectile on mild steel plates. International Journal of Impact Engineering, 12(3), 333–343. https://doi.org/10.1016/0734-743X(92)90101-X
Johnson, W., Sengupta, A. K., & Ghosh, S. K. (1982). High velocity oblique impact and ricochet mainly of long rod projectiles: An overview. International Journal of Mechanical Sciences, 24(7), 425–436. https://doi.org/10.1016/0020-7403(82)90052-2
Lee, S. W. R., & Sun, C. T. (1993). Dynamic penetration of graphite/epoxy laminates impacted by a blunt-ended projectile. Composites Science and Technology, 49(4), 369–380. https://doi.org/10.1016/0266-3538(93)90069-S
Shen, C.-H., & Springer, G. S. (1976). Moisture Absorption and Desorption of Composite Materials. Journal of Composite Materials, 10(1), 2–20. https://doi.org/10.1177/002199837601000101
Reviewer 3 Report
This paper presnts a study about the influence o hygrothermal aging in CFRP materials under high velocity impacts. Despite the paper is clearly presemted there are some deficiencies in the presentation and discussion of the results. Some guidelines ara mentioned next:
- How do you fix the specimens? Did you use any special device? In this point is important to have into account the influence of the screws, in case you used, as stress concentrators.
- How many timed did you repeat each test?
- In figure 5 a high variablity can be observed in some test whereas in other the variability is almost 0. What is the reason of this? At the same time, the varibiltity is so high that hat the trending is not representative. In somecases you only present one point (flat, wet case) what has no sense in my opinon. Residual velocity can be measured as well, I do not know why you did not collect it. In my opinion, 1 or 2 points for each case are not enough to oberve trendings of a specific behaviour, You sould try to get more data in the invertal.
- Does the ballistic limit change with the moisture absorption? Did you calculate it?
- How do you quantify the damage? what equation/process do you use? IN figure 12 the second point of flat/wet case presents a high error. What is the reason of this?
- Did you think about the influece of the time (10 days, for example) after you saturate a sample? does the moisure absoprtion decreases with the time? This point is important because if the material is used in service and properties decrease with time because of the loss of moisure, the piece can fail.
- Did you observe repeatability of the failure method in test carry out under the same conditions? What about the shape of damaged area?
- The scale of the figures is to small that sometimes you can not even read it. Please make them bigger.
- There is a mistake in line 274. The las word is "th e" -> "the"
Author Response
Reply to Questions of Reviewer #3
General comments and Suggestions for Authors:
This paper presents a study about the influence of hygrothermal aging in CFRP materials under high velocity impacts. Despite the paper is clearly presented there are some deficiencies in the presentation and discussion of the results. Some guidelines are mentioned next:
A: Firstly, we would like to thank the referee for spending much time reviewing our manuscript to materials. Based on the reviewer’ useful comments, we have seriously revised the paper. In this response letter, we have provided a written reply (below) to all the questions raised by the referee. If the referee has further questions, please let us know. We shall further work on them.
Q1- How do you fix the specimens? Did you use any special device? In this point is important to have into account the influence of the screws, in case you used, as stress concentrators.
Re: Thanks for carefully reading this manuscript and providing the kind comments. In our study, the sample is fixed like a special-designed simply support boundary, instead of fastened on the fixture by strews. The detailed information of the fixture is shown below.
Figure R1 Schema of the clamping device.
The CFRP plates are clamped on the base fixture by a cover plate which has a square hole. The CFRP plate has a dimension of 115mm×115mm. The central hole has a dimension of 100mm×100mm. Total clamping device is supported by another holder. There are no stress concentrations on the CFRP plate caused by the screws in our test as shown in Fig. R1. To avoid misunderstanding, the corresponding figure has been corrected. Please refer to Figure 2.
Q2- How many times did you repeat each test?
Re:Thanks very much for the comments. For each test, we repeated it 8-12 times for either the “wet” specimen or the “dry” specimen at each velocity with each kind of projectile.
Q3- In figure 5 a high variability can be observed in some test whereas in other the variability is almost 0. What is the reason of this? At the same time, the variability is so high that hat the trending is not representative. In some cases you only present one point (flat, wet case) what has no sense in my opinion. Residual velocity can be measured as well, I do not know why you did not collect it. In my opinion, 1 or 2 points for each case are not enough to observe trending of a specific behavior, you could try to get more data in the interval.
Re: We would like to thank the referee for reading this manuscript carefully and providing the comments.
Firstly, we are sorry that the incorrect figure has been inserted into the submitted paper. Now the figure 5 has been replaced with the correct one after carefully checking. Please refer to Figure 5.
Secondly, much thanks to the referee for pointing out the high variability. It should be noticed that the high-velocity impact test still has some uncontrollable factors even though they are almost successfully completed. The first one is the imperfect controlling of the incident angle, which is the dominated factor, since the projectile is very sensitive to the incipient posture (Gupta & Madhu, 1992). Slight error in the incident angle may deviate the after-penetration or ricochet path (Johnson, Sengupta, & Ghosh, 1982), owing to the moment formulated by the contact force and the eccentric distance. Particularly for the blunt-ended projectile, there even exists additional influence from the air resistance. Obviously, an oblique route leads to more energy loss and longer path. The second one should be noted is the variation of the impact velocity. Owing to that the velocity of projectile is controlled by the nitrogen pressure, the impact velocities are not exactly the same for the cases with specified velocity and projectile. The third one is due to the processing variability of the specimen. It would lead to the local discrepancies in the porosity and matrix content, and further introduce discrepancy of impact response. The last one is the mechanical error of gas gun system, such as the pressure and the updated posture of the clamping device after each impact. It is right that single factor may have slight influence, so that most of the data are convergent. With the combine influence of those factors, a few scattered data were found in some cases, which showed an evident standard deviation.
Thirdly, thanks for the comments. We need to explain that the missing data was for the cases where the rebounding of the projectile happened. To the best knowledge of the author, the effective techniques of measuring the rebounding velocity are very complex. It was difficult using the existed devices to measure the rebounding velocity both accurately and effectively in this study. The route of the rebounding was not always normal to the laminate, and the induced oblique ricochet really limited the effective measurement. Since the data in those (ricochet) cases are not effective and inadequate, their data of the kinetic energy loss were not given, e.g. for the cases of the flat- and sphere- ended projectile.
Whereas it should be noted in the current studies that, the result of single test includes both the perfect conditioned results and the poor one. The final result for each test is the combination of them. Since the result with the high variability still reflects a law although not very strong, it could support our trending analysis and the further conclusion. As each test has been repeated many times, the existed results should be right for the mentioned velocity. Thus, it could be understood that such trending analysis is right whereas limited, so that the conclusion has been reorganized as the trending is effective within the limited velocity range. Please refer to lines 354-356.
Finally, it is very kind of the referee to propose the supplementary data to improve the work. It is really significant and meaningful to supplement additional experiments, and we will definitely implement the tests in the future work according to the comments, since the hygrothermal experiment really cost a long time in both the specimen preparation and saturation. Sincerely thanks for the suggestion and we will implement this experiment in the future.
Q4- Does the ballistic limit change with the moisture absorption? Did you calculate it?
Re: Thanks for the comments. Yes, the ballistic limit will change with moisture absorption according to the prediction derived from our results.
The referee provides a valuable illumination for us to improve this work. The calculated results have been given in a separate paper. Since this study concerns more about the effect of hygrothermal conditioning on the energy absorption and the failure mechanism under high-velocity impact, we have not included the ballistic limit in this study. The corresponding further analyses may be out of the range of the concern in the present study.
Q5- How do you quantify the damage? What equation/process do you use? IN figure 12 the second point of flat/wet case presents a high error. What is the reason of this?
Re: Thanks very much for the comments.
Firstly, the damaged was quantified using the c-scanning device (UPK-T48-HS, PAC Inc.) with 10 Hz flat transducer. Please refer to lines 115-116.
Secondly, after the scanning, the delamination could be further post-processed on the UTwinTM software, and the averaged projected area is automatically calculated. Please refer to lines 116-117.
Thirdly, regarding the case of the specimens impacted by flat-ended projectile at velocity 68 m/s, the high error may be induced by the combination of two factors. The first one is the particular shape of the projectile. With greater area of the end, the aerodynamic force upon the projectile is more evident than other projectile. Such blunt ended projectile is sensitive to the incident angles (mentioned in Re3), the potential oblique posture is irregular so that the projected perforated area is different. Particularly when in the high velocity impact, the influence will also be evident with the special failure mode of the wet specimen, whose fiber and matrix display less resistance. Along with the additional influence from the air resistance, the oblique penetration became more complicated or even missed the target. The second one may be the specimen (mentioned in Re3). Since local discrepancy of the porosity and matrix extent in the specimens may also influence the damages to different extent. As a summary, the second point of flat/wet case may have a few scattered data with evident standard deviation.
Q6- Did you think about the influence of the time (10 days, for example) after you saturate a sample? Does the moisture absorption decreases with the time? This point is important because if the material is used in service and properties decrease with time because of the loss of moisture, the piece can fail.
Re: Thanks for providing kind comments.
We have considered such influences during the test. And the hygrothermally conditioned laminates were tested on the light-gas gun loading system in 2 hours after it was been exposing to the air. During this period, no evident decrease of the moisture absorption of the saturated specimen was found in our test.
When the samples are dehydrated for a long time, significant loss of the moisture may be observed. However, it is out of the range of the present study. An accurate and adequate explanation is difficult to be given in this work, maybe a further specific analysis in the future work could be carried out to make it clear.
Lastly, thanks for the comments about analyzing the significance of dehydration after hydrogenthermal condition for us. We will carefully notice those experimental details in our future work. Also, it enlighten us with a significant issue and provide a kind idea to improve the present work,
Q7- Did you observe repeatability of the failure method in test carry out under the same conditions? What about the shape of damaged area?
Re: Thanks for the referee giving the kind comments.
Yes, we observed the repeatability of the failure under the same conditions. Most damaged area shape of the specimens under the impact conditions suggest such a law, while the damaged shape presented in the study could be more representative. For example, we could provide another damage areas for the case of impact velocity 88 m/s and flat projectile, as shown in Figure R2. In this figure, it also suggests that the shape of the damaged area is rhomboidal for the dry specimen, and spherical for the saturated specimen. As a result, we could predict such shape repeatability has a law.
Figure R2. Damaged areas of the cases with impact velocity of 88 m/s and flat projectile
Q8- The scale of the figures is too small that sometimes you cannot even read it. Please make them bigger.
Re:Thanks very much for reading the manuscript carefully. We have increased the sizes of Figures 13-16.
Q9- There is a mistake in line 274. The last word is "th e" -> "the"
Re:Thanks very much for pointing out this. We have corrected it. Please refer line 314.
Reference
Gupta, N. K., & Madhu, V. (1992). Normal and oblique impact of a kinetic energy projectile on mild steel plates. International Journal of Impact Engineering, 12(3), 333–343. https://doi.org/10.1016/0734-743X(92)90101-X
Johnson, W., Sengupta, A. K., & Ghosh, S. K. (1982). High velocity oblique impact and ricochet mainly of long rod projectiles: An overview. International Journal of Mechanical Sciences, 24(7), 425–436. https://doi.org/10.1016/0020-7403(82)90052-2
Reviewer 4 Report
The authors studied the influence of hygrothermal aging on high-velocity impact damage of CFRP laminates. They used three types of impact shapes and velocities. The impact induced damages (delamination) is observed and quantified by ultrasonic C-scan, digital microscope system and scanning electron microscope. The work is well structured and the conclusions are supported by the results. The results are well documented with numerous images and are quite clear. The work should be accepted. Only some clarifications:
1) Did you tried other stacking sequences different to [0,90]? Please, considere it for future research.
2) Did you tried to use acoustic emission recording during impact test? There are some papers about it. Please, considere it for future research.
3) Figure 5. Why there are only one velocity point for flat wet shape?
4) Figures 12 and 5. Why there are 3 velocity points at Figure 12 but only 2 at Figure 5?
Author Response
Reply to Questions of Reviewer #4
General comments and Suggestions for Authors:
The authors studied the influence of hygrothermal aging on high-velocity impact damage of CFRP laminates. They used three types of impact shapes and velocities. The impact induced damages (delamination) is observed and quantified by ultrasonic C-scan, digital microscope system and scanning electron microscope. The work is well structured and the conclusions are supported by the results. The results are well documented with numerous images and are quite clear. The work should be accepted. Only some clarifications:
A: Firstly, we would like to thank the referee for spending much time reviewing our manuscript to materials. Based on the reviewer’ useful comments, we have seriously revised the paper. In this response letter, we have provided a written reply (below) to all the questions raised by the referee. If the referee has further questions, please let us know. We shall further work on them.
Q-1 Did you tried other stacking sequences different to [0,90]? Please, consider it for future research.
Re: Thanks very much for carefully reading this manuscript. Actually, we have never tested the other stacking sequences different to [0, 90]. Thanks for the kind suggestions, and we will consider it in our future work according to the comments.
Q-2 Did you tried to use acoustic emission recording during impact test? There are some papers about it. Please, consider it for future research.
Re: Much thanks to the referee for providing the kind suggestions. Acoustic emission recording is not used in the current study, whereas it is a significant technique and can provide effective measurement in the impact test. According to the comments, we would like to conduct the related studies, consider the technique and employ it in our future research.
Q-3 Figure 5. Why there are only one velocity point for flat wet shape?
Re: Thanks very much to the referee for pointing out this. This figure has been improperly inserted into the paper during the submission. It has been corrected. Please refer to Figure 5 in the revised paper.
Q-4 Figures 12 and 5. Why there are 3 velocity points at Figure 12 but only 2 at Figure 5?
Re:Thanks for the referee pointing out this. In Figure 5, the energy absorption equals to the energy loss and is calculated using the residual velocity. Since some cases in low velocity did not perforate the specimen and ricochet, their residual velocities are not effectively measured in the current study. As the ricochet of the projectile is sensitive to the incident angle, which is difficult to be controlled in the experiment, the ricochet path could not be accurately and effectively recorded, and the measured data is inadequate. So, their energy loss is not given. And there are only 2 velocity points at Figure 5. Instead, the damaged area at Figure 12 is not related with the residual velocity and could be computed by the c-scan device, so that the 3 velocity points could be simply given.
Round 2
Reviewer 3 Report
Questions 1,2,4,6,7,8 and 9 are responsed correctly.
-Answer of Q2 should be included in the paper, please.
-in Q3 the figure was modified. In this point, to have a better data you should not include the poor one data. Despite that, at least 3 point should be included in the grapich to define a trending. If it is possible I insist on include, at least, ine more point for the sphere and the flat projectile.
-Answer for Q5 justify the high error of the Figure 12 in the second point of flat/wet case is because the projectile shape and the specimen. Following the same reasoning, the error for the other 2 points should be also high but it is not. Why? the same case occurs with the third point of flat/dry case. The reason could be that you are evaluating some repetitions that are not correct and they are increasing the standard deviation. The point is that the dispersion should be similar for the three cases.
Author Response
Reply to Questions of Reviewer #3
Q1- Questions 1,2,4,6,7,8 and 9 are responsed correctly.
Re: Firstly, we would like to express many thanks to the referee for spending much time reading these response and providing the kind comments.
Q2- Answer of Q2 should be included in the paper, please.
Re:Thanks very much for the comments. We have included the answer in the paper. Please refer to lines 11-12.
Q3- in Q3 the figure was modified. In this point, to have a better data you should not include the poor one data. Despite that, at least 3 point should be included in the grapich to define a trending. If it is possible I insist on include, at least, ine more point for the sphere and the flat projectile.
Re: Thanks very much for the comments.
Firstly, according to the comments of the referee, we have excluded the poor data which led to evident standard deviation in the previous Figure 5. The data and the corresponding standard deviation have been recalculated, including the wet specimen impacted at velocity 68 m/s by the flat and sphere projectiles, the wet specimen impacted at velocity 88 m/s by the sphere projectile, and the dry specimen impacted at velocity 88 m/s by the cone projectile. The standard deviation for the same projectile kind are similar with an insignificant variation, such as the wet/dry specimen impacted at velocity at velocities 68 m/s and 88 m/s by the flat projectile, as shown in Figure R1.
Figure R1. Recalculation of the kinetic energy loss by excluding the poor data according to the comments. (Current Figure 5 in the revised paper)
It is shown that the previous discussion of the trending analysis is also supported by the updated data. Because of the recalculation, the corresponding results and marked values of the comparison have been updated in the revised paper. Please refer to lines 168-169, 179-180 and Figure 5.
Secondly, following the referee’s useful suggestions, we have provided the data obtained in each of the tests with velocity of 45 m/s and the sphere/flat projectiles. Although those data are inadequate and may be improper for the quantitative analysis, the referee is right that the addition of those data could be helpful to the trending analysis. Because of the mentioned difficulty in the accurate measurement of rebounding velocity, only one data was collected for those tests involving ricochet so that there is no bar to mention the standard deviation for them. Please refer to lines 152, 153, 158-159.
Q4- Answer for Q5 justify the high error of the Figure 12 in the second point of flat/wet case is because the projectile shape and the specimen. Following the same reasoning, the error for the other 2 points should be also high but it is not. Why? the same case occurs with the third point of flat/dry case. The reason could be that you are evaluating some repetitions that are not correct and they are increasing the standard deviation. The point is that the dispersion should be similar for the three cases.
Re: We would like to thank the referee for analyzing the high deviation in the results for us and providing the kind suggestion.
The referee is right that some repetitions are incorrect and the poor data really leads to the evident standard deviation. According to the comments, we have removed the incorrect data, and the damaged area for these cases has been recalculated, including the wet specimen impacted at velocity 68 m/s by the flat projectile, the dry specimen impacted at velocity 88 m/s by the flat projectile, the wet specimen impacted at velocity 88 m/s by the sphere projectile, as shown in Figure R2.
Figure R2. Recalculation of the damaged area by excluding the poor data according to the comments. (Current Figure 12 in the revised paper)
It is also shown that the previous analysis of the trending is still supported by the updated data. The corresponding results and marked values of the comparison mentioned in the revised paper have been also updated. Please refer to lines 256-257, 265 and Figure 12.